# An Analysis of Mechanical Properties for Ultrasonically Welded Multiple C1220-Al1050 Layers

**Jisun Kim [1], Jaewoong Kim [1,*], In-ju Kim [1], Sungwook Kang [2]**  **and Kwangsan Chun [3]**

[1] Smart Manufacturing Process R&D Group, Korea Institute of Industrial Technology, Gwangju 61012, Korea; kimjisun@kitech.re.kr (J.K.); k9inju@kitech.re.kr (I.-j.K.)

[2] Transport Machine Components R&D Group, Korea Institute of Industrial Technology, Jinju-si 52845, Korea; swkang@kitech.re.kr

[3] Welding Engineering R&D Department, Industrial Application R&D Institute, Daewoo Shipbuilding & Marine Engineering Co., LTD, Geoje-si 53302, Korea; kschun@dsme.co.kr

* Correspondence: kjw0607@kitech.re.kr; Tel.: +82-62-600-6480

**Abstract:** This study analyzed the characteristics of aluminum and copper sheets under multi-layer ultrasonic welding, and observed the strength, fracture type, and interface of the weld zone according to location. In addition, an experimental plan was developed using the Taguchi method to optimize the quadruple lap ultrasonic welding process conditions of 0.4t aluminum and copper sheets, and the experiment was performed for each of 25 welding condition. For strength evaluation, the ultrasonic welding performance was evaluated by measuring the tensile strength as a composite material and the shear force at the weld interface through two types of tensile tests: simultaneous tensile and individual tensile. To identify the individual shear strengths of the multi-layer dissimilar ultrasonic welds, three types of tensile tests were performed for each specimen depending on the location of the welded, and as the distance from the horn increased, each of shear strength decreased while the difference in strength value increased. For quadruple lap welding of pure aluminum and copper sheets, the S/N (Signal to Noise Ratio) was the highest at 64.48 with a coarse-grain pattern and optimal welding conditions, and this was selected as the optimal condition. To evaluate the optimized welding condition, additional tests were conducted using the welding conditions that showed the maximum strength values and the welding conditions optimized using the Taguchi method through simple tests. A strength evaluation of the optimized weldment was performed, and for a simultaneous tensile test, it was found that the strength of the optimized weldment was improved by 45% compared to other cases.

**Keywords:** ultrasonic welding; optimization; tensile strength; shear strength; aluminum sheet; copper sheet; Taguchi method

## 1. Introduction

Ultrasonic welding is one of the representative welding technologies using plastic flow, with low electrical resistance as the oxide film and impurities are removed and welded through the diffusion of metal by vibration [1–3]. Ultrasonic welding also makes it possible to weld nonferrous metals or dissimilar metals that are difficult to bond using conventional methods, and it is mainly used for thin foils, because its effect on metal is relatively small compared to other welding methods due to the very short welding time. Research has found that ultrasonic metal welding is an excellent precision welding method, as it is very much superior in terms of process time and contact resistance [4].

In addition, ultrasonic welding makes it easy to weld the same kind of metals or dissimilar metals which are difficult to weld using conventional methods; additionally, since the weld can be performed

at a low temperature, it minimizes thermal damage due to low thermal expansion. It also has been capturing attention as a bonding technology for high-performance, multi-functional parts, because it is an eco-friendly method that does not use solder [5].

The dissimilar welding of aluminum and copper uses non-fusion bonding technologies, such as ultrasonic welding, electromagnetic pulse welding, and friction stir welding [6,7]. For fusion welding, a strong, brittle intermetallic compound (AlxCuy) is formed at the joint, which reduces the bonding performance [8–10]. In addition, these intermetallic compounds are reported to reduce the electrical conductivity. For these reasons, bonding methods that use mechanical bonding or plastic flow are used for these two metals, which have low melting properties [11–15].

Ultrasonic welding has recently been applied to bonding nickel and aluminum in automobile battery cells or for the micro-wire bonding of electronic products [16,17]. However, ultrasonic welding technology is now being applied even to bonding tabs and bus bars to modularize secondary battery packs for vehicles. The bonding of tabs and bus bars in secondary batteries is a leading example of aluminum and copper bonding [18,19]. Lately, ultrasonic welding has been applied to connecting a terminal with CCAW (copper-clad aluminum wire), which has been developed to reduce the weight and improve the price-competitiveness of copper wires [20]. CCAW is composed of an inner aluminum core and outer copper cladding, processed by continuous rolling to within 0.01 mm, and attached to an aluminum or copper terminal [21]. This CCAW welding process is part of the dissimilar metal multi-layer bonding process.

The purpose of this study is to analyze the strength characteristics of the multi-layer welding process of dissimilar metal (aluminum and copper) sheets and to determine the optimal welding conditions. To analyze the strength characteristics of the multi-layer welds, individual tensile tests were conducted for each layer and the effects of the welding conditions on the strength of a specific location were analyzed. In addition, the Taguchi experimental design method was used for optimization to increase the strength of each weld zone.

## 2. Experimental Procedure

### 2.1. Experimental Composition and Material

Ultrasonic welding, as shown in Figure 1, is a process in which high-frequency vibrations of 20 kHz or higher are partially applied to the welding material to form a joint. When ultrasonic waves are used for welding metallic materials, they are applied in the lateral direction parallel to the material's surface, and when they are used for welding plastic materials, they are mainly applied in the longitudinal direction perpendicular to the material's surface [1,2]. The ultrasonic metal welding system consists of a power supply, converter, booster, and horn, and by converting the electrical energy of 50–60 Hz coming from the power supply into mechanical vibration energy of 20–40 kHz through the converter, when the booster amplifies the micro-vibrations to be used in metal welding, the oxide film and the contaminated film on the metal surface are destroyed by the frictional heat generated by the vibration on the deposited metal through the horn, thus forming a metallic bond [22,23].

The ultrasonic welder used in the experiment, as shown in Figure 2, was a Komax 2050 model with a frequency characteristic of 20 kHz. There are three major variations in welding conditions through the controller, which are the welding time, welding force, and ultrasonic wave output strength. Two non-ferrous materials were prepared for this experiment, which are industrial grade pure metal (purity 99.0%–99.9%) 1000 series aluminum non-heat treatable alloy, and phosphorus deoxidized copper. The 1000 series aluminum alloys have a relatively weaker mechanical strength than other aluminum alloys, but are mainly used as wrought alloy because they have good corrosion resistance, light reflectivity, electrical and thermal conductivity, and excellent workability and weldability. The phosphorus deoxidized copper contains 0.015%–0.040% residual phosphorus (P), shows a uniform microstructure compared to oxygen-free copper, and has excellent ductility and fatigue strength and

good weldability. The mechanical properties of the materials used in the experiment are shown in Table 1.

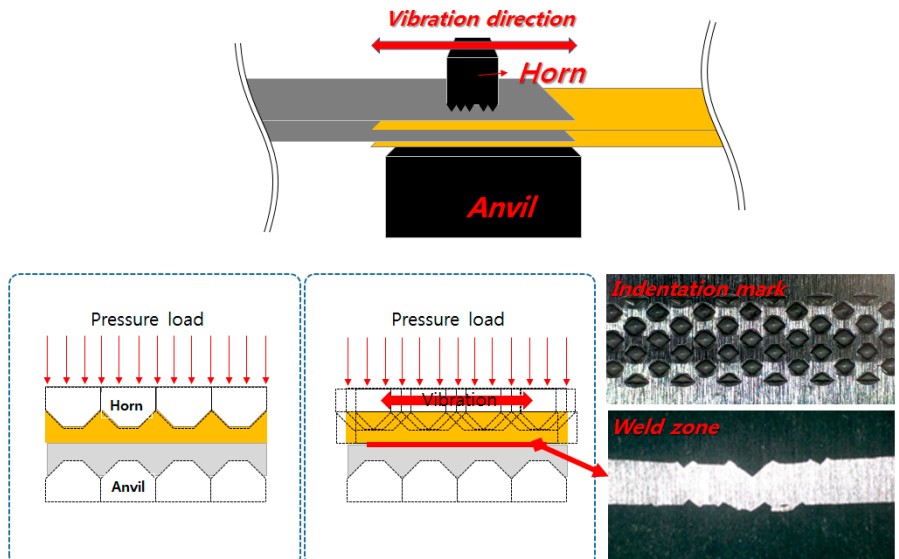

**Figure 1.** Ultrasonic welding process.

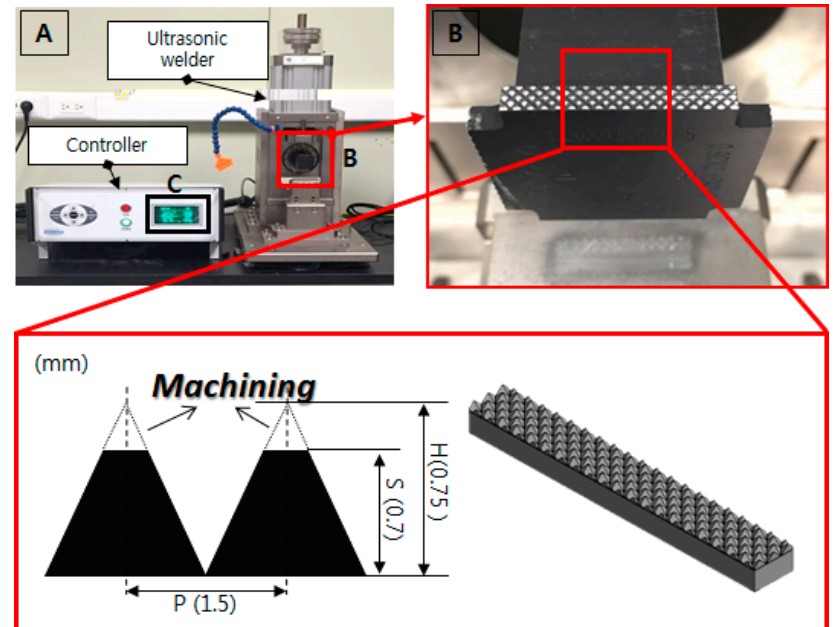

**Figure 2.** Ultrasonic metal welding systems employed in experiment. (**a**) Systems configuration; (**b**) pattern shape on horn.

**Table 1.** Mechanical properties of base metal.

| Material | Density (g/cm³) | Modulus of Elasticity (GPa) | Poisson's Ratio | Ultimate Tensile Strength (MPa) | Yield Strength (MPa) |
|---|---|---|---|---|---|
| Al (A1050) | 2.71 | 68 | 0.30 | 138.3 | 125.5 |
| Cu (C1220) | 8.94 | 117 | 0.38 | 266 | 246 |

In this study, 20 mm × 100 mm × 0.4 t specimens, as shown in Figure 3, were used for lap joint welding; these specimens were also used to evaluate the strength of weld after the experiment was

completed. The experiment was performed in two groups with a total of 25 welding conditions. The base materials used in the experiment were aluminum 1050 series and copper 1220 series. A total of four layers were formed for multi-layer dissimilar metal welding, and the aluminum and copper were arranged to cross each other. There are two types of specimens produced through the experiment. As shown in Figure 3a, the experiment with four specimens stacked up in a uniform arrangement was designed to simultaneously tension the four specimens, and in Figure 3b, the specimen was prepared to measure the shear strength by level in order to obtain the strength of each weld individually. A total of (25 × 2 = 50) experiments were conducted twice using 25 experimental conditions. The surfaces to be welded were cleaned in the order of stainless wire-brush, then sand paper—#400, #800, and #1500, to prevent foreign substances such as rust, scale, and oxides from causing weld defects.

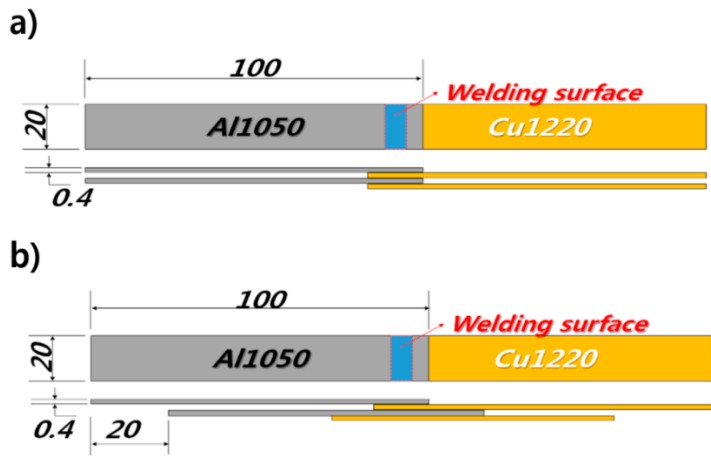

**Figure 3.** Configuration of welding specimen.

## 2.2. Design of the Experiment

For this study, the experiment used the Taguchi experiment plan. The Taguchi method is an experimental planning method which is effective it making a large number of decisions through a small number of experiments [24,25]. This method uses a table of orthogonal arrays and S/N (signal-to-noise ratio) to determine the factors that have a significant effect on the experimental results and to obtain the optimal conditions. The specific value in the Taguchi method refers to the result value of a typical welding experiment. The characteristic value in this experiment was selected as the strength of the weld. In general, the most important aspect when evaluating a weld is its strength. Therefore, the characteristic value was obtained in this study by measuring the shear tensile strength. Noise is a factor that affects the characteristic value, and it is referred to as a welding condition in the field of welding. The typical welding conditions in ultrasonic welding are welding time, welding force, and amplitude. To set the levels of the three major variables of ultrasonic welding, which are welding time, welding force, and amplitude, preliminary experiments were conducted to determine the welding range, and the level of the experimental table was prepared using the table of orthogonal arrays. While many levels are required to secure the reliability of the experiments' results, it is necessary to select an appropriate level of confidence by the appropriate user, as the number of experiments increases as the levels increase. Through preliminary experiments, it was determined that if welding is performed for more than 5 seconds, the life of the horn is reduced due to the overload; amplitude was also planned to be 100% or less. For the welding force, it is possible to pressurize at a maximum pressure of 6.3 bar and a minimum of 0.5 bar. However, it was impossible to weld the metal and conduct the shear strength test at 0.5 bar. After selecting the level of the factors, the table of orthogonal arrays for the experiment must be selected. The table of orthogonal arrays is used to perform experiments by arranging the selected factors and identifying the effect of each factor, in order to obtain the maximum effect with minimum experiments. The table of orthogonal arrays used in this study is L25 (53), a 3-factor table

with 5 levels that can perform a total of 25 experiments. In this experiment, the levels shown in Table 2 were determined to establish the experimental plans.

**Table 2.** Welding parameters and their levels for Taguchi method.

| Material | Level | | | | |
|---|---|---|---|---|---|
| | 0 | 1 | 2 | 3 | 4 |
| Amplitude rate (%) | 40 | 50 | 60 | 70 | 80 |
| Welding time (s) | 2 | 2.5 | 3 | 3.5 | 4 |
| Pressure load (bar) | 1 | 1.5 | 2 | 2.5 | 3 |

## 3. Results of the Experiment

### 3.1. Evaluation of Mechanical Joint Performance

The specific value examined in this study is the shear strength. Two types of specimens were used to measure the shear strength, and the data obtained from each specimen are as follows. (1) Maximum tensile strength measurement using a test method that tensions the entire multi-layer welding specimen. (2) Shear strength measurement of three welds by tensioning each weld zone. A 10 ton UTM (Universal Testing Machine) was used to measure the shear strength, and to measure the maximum tensile strength, cover plates were inserted between the base materials, as shown in Figure 4, so that the load was transferred horizontally to measure the precise shear strength. Figure 5 shows the experimental procedure for measuring the multi-step shear strength of the weld. To measure the shear strength of each weld, the experiments were conducted in a sequence: Al–Cu tensile test, Cu–Al tensile test, and Al–Cu tensile test. Since there are no standards for measuring the shear strength of ultrasonic welds, a comparison of the welding conditions was performed by forming welds of the same area and measuring the quantitative shear strength.

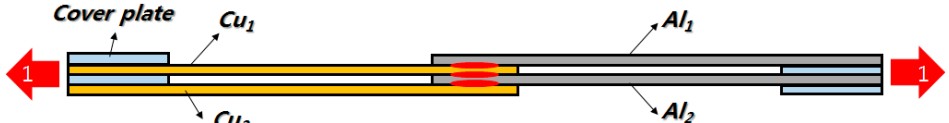

**Figure 4.** Procedure of one-step shear strength test for multi-lap joint welding.

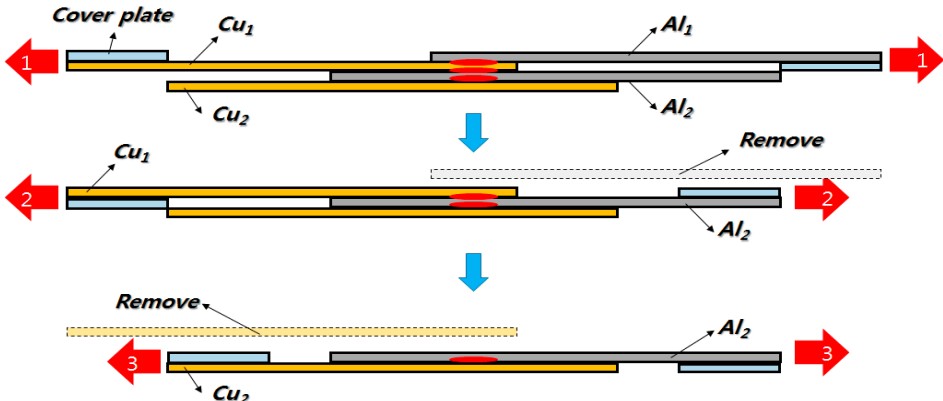

**Figure 5.** Procedure of multi-step shear strength test.

The shear strength measurement test results using a UTM are divided into three types through the displacement–load curve. As shown in Figure 6, three interfaces are created when ultrasonic welding with four sheets, and their characteristics are as follows. Three ultrasonic welds are used to weld the four thin plates.

1)    Weld zone 1: Interface between aluminum (A1) and copper (C1) (aluminum weld indentation);
2)    Weld zone 2: Interface between copper (C1) and aluminum (A2);
3)    Weld zone 3: Interface between aluminum (A2) and copper (C2).

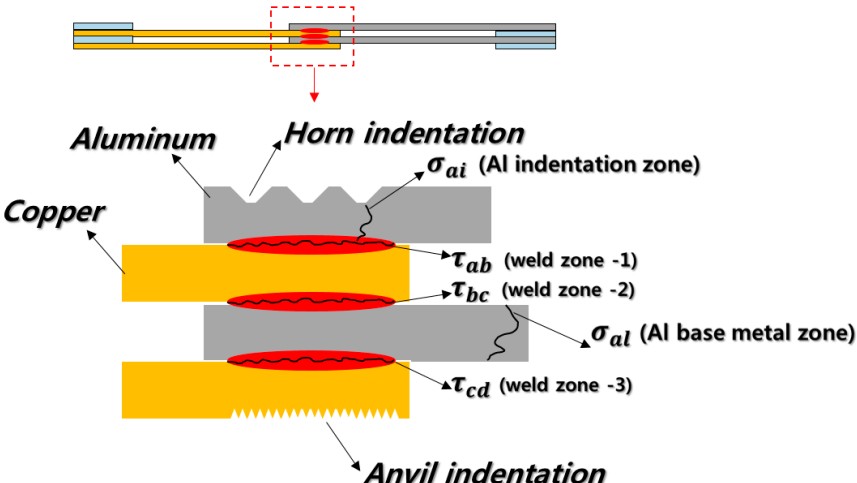

**Figure 6.** Three zones of the ultrasonic weldment of multi-layer, dissimilar materials Al and Cu.

Weld zone 1 in Figure 6 is bonded together with the aluminum and copper surface, and the indentation of the horn reduces the sectional area of the aluminum weld. The fracture location in weld zone 1 is divided into an interface fracture and an indentation fracture, and it has the form of an aluminum fracture because the strength of aluminum is weaker than that of copper. For weld zone 2, the fracture is located at the center of the weld in the form of a joint between the upper copper plate and the lower aluminum plate. It also shows the same interface fracture or aluminum fracture as weld zone 1. Weld zone 3 is the same as weld zone 2. The aluminum plate that exists in both weld zone 2 and 3 bears the form of a base material fracture when weld zone 2 and 3 are stably bonded.

The shape of the welded specimen is shown Figure 7a. At the cross section in Figure 7b, the bond line was not observed due to the characteristics of solid phase bonding. When resistance welding, the bond line is observed but both heat affected zone and bonding line are not observed, since ultrasonic metal welding is one of the plastic-flow welding methods [26,27]. However, it was confirmed that two metals were closely bonded to each other after observing the weldment under 3000 times magnification, as shown Figure 7c.

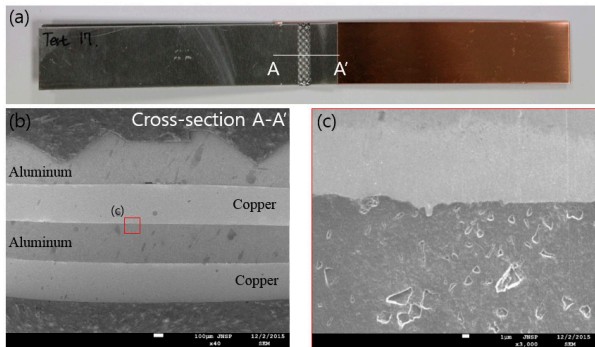

**Figure 7.** Cross-section ultrasonic weld zone with amplitude 70%, welding time 2.5 sec, and pressure load 3 bar: (**a**) cross-sectional cutting position; (**b**) joint cross section (x40); (**c**) joint cross section (x3000).

### 3.1.1. Evaluation of Mechanical Joint Performance

When the multi-layer ultrasonic welds are simultaneously tensioned, the fracture type can occur in five different shapes, as shown in Figure 8. Figure 8a is a case in which interface fracturing occurred in all of the welds. In case b, the break occurs at the end of the indentation, and then the interface fracture occurs in all of the welds. In case c, an indentation fracture occurs in weld zone 1 and an interface fracture occurs in weld zone 2. In addition, a fracture occurs in the aluminum base metal between the copper plates. In such a case, the shear strength is similar to the tensile strength of the aluminum base material. In case d, the primary fracture occurs in the indentation of the upper aluminum, and both interface and aluminum fractures coexist. In cases c and d, fracture types of aluminum—indentation fracture, interface fracture, and aluminum base material fracture—exist in the same way, but the location of the interface fracture is different. In case e, only two fracture types are shown. In this case, the second aluminum sheet is fractured in a state where the aluminum indentation fracture type of weld zone 1, and weld zone 2 and 3 are bonded together. However, when the welds were tensioned separately, a total of four types of fracture were observed, similar to the case of simultaneous tensioning. In Figure 9a, interface fractures occurred in all welds, and in case b, the aluminum base material fracture occurred in Al1–Cu1. Subsequently, interface fractures occurred in all of the other welds. In case c, aluminum indentation fracture occurred in Al1–Cu1, and aluminum base material fracture occurred in the Cu1–Al2 weld. In case d, aluminum fracture occurred because of indentation in the Al1–Cu1 weld, and interface fracture occurred in weld zones 2 and 3. Observing the welds separately, aluminum base material fractures, aluminum indentation fractures, and interface fractures were all observed in weld zone 1, and aluminum base material fractures and interface fractures occurred in weld zone 2. For weld zone 3, no type of fracture other than interface fracturing was observed.

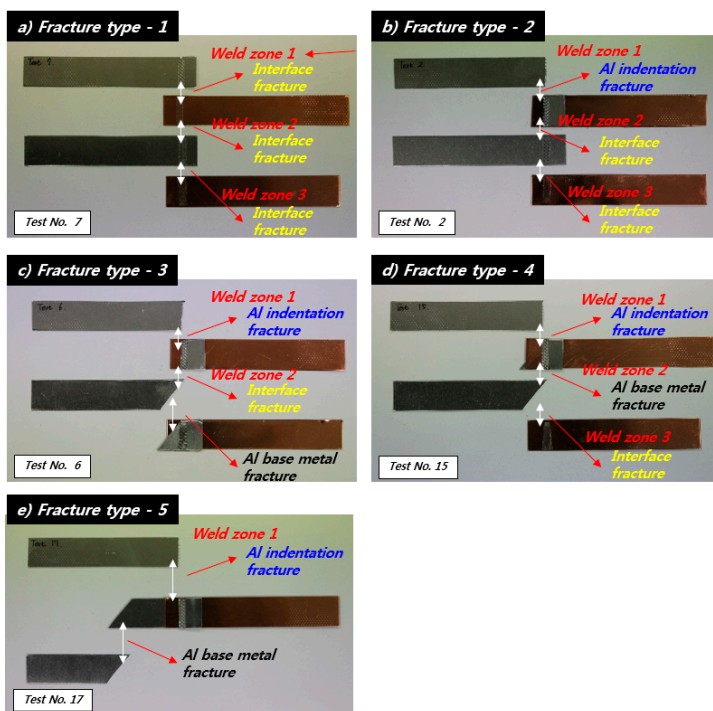

**Figure 8.** Five kinds of fracture types in a one-step shear strength test. (**a**) Interface fracture at all weld zones, (**b**) Combination fracture of Al indentation and interface, (**c**,**d**) Combination fracture of Al indentation, interface and Al base metal, (**e**) Combination fracture of Al indentation and Al base metal.

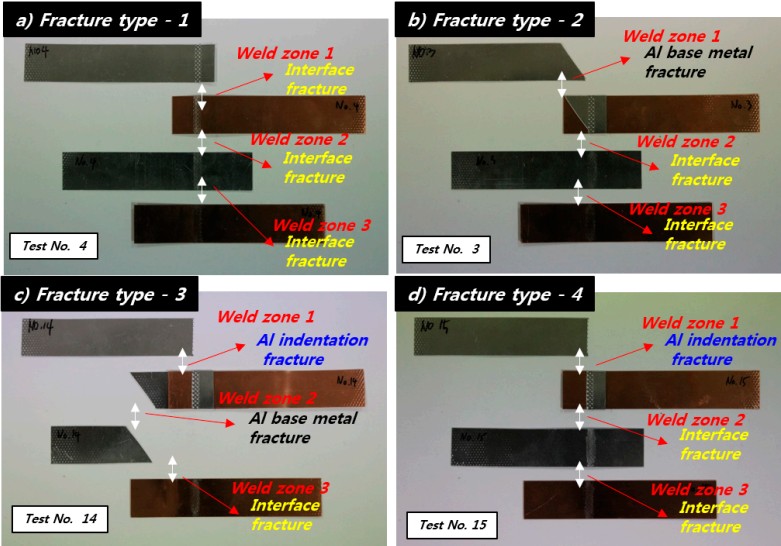

**Figure 9.** Four kinds of fracture types in a multi-step shear strength test. (**a**) Interface fracture at all weld zones, (**b**) Combination fracture of Al indentation and interface, (**c**) Combination fracture of Al indentation, Al base metal and interface, (**d**) Combination fracture of Al indentation and Al base metal.

### 3.1.2. Fracture Sequence

Ultrasonic multi-layer welding has three welds, and even if they are simultaneously tensioned to measure shear strength, they cannot be fractured at same time. The difference in strength between each weld and the material causes sequential fracture and the fracture location of the weld in the form of stress is shown as Figure 6. Because stress is related to the weldability, the better the weldability and the greater the force to withstand external forces that can be guaranteed by higher stress. Four fracture sequences were inferred through this experiment, and the fracture sequence, as shown in Table 3, could be predicted by the differences of stress at the fracture points. In Figure 6, $\sigma_{ai}$ is tensile stress of indentation at the aluminum upper sheet and σ_al is shear stress in the aluminum base material. The shear stresses of weld zones from 1 to 3 are $\tau_{ab}$, $\tau_{bc}$, and $\tau_{cd}$.

**Table 3.** Four kinds of fracture type.

| Case | Stress Condition | Fracture Sequence | | |
|------|------------------|-------|--------|-------|
| | | First | Second | Third |
| 1 | $\sigma_{ai} > \sigma_{al} \geq \tau_{ab} \cong \tau_{bc} \cong \tau_{cd}$ | Weld zone 3 | Weld zone 2 | Weld zone 1 |
| 2 | $\sigma_{al} > \tau_{ab} \cong \tau_{bc} \cong \tau_{cd} > \sigma_{ai}$ | Al indentation zone | Weld zone 2 | Weld zone 3 |
| 3 | $\tau_{ab} \cong \tau_{cd} > \sigma_{al} > \tau_{bc} > \sigma_{ai}$ | Al indentation zone | Weld zone 2 | Al base metal |
| 4 | $\tau_{ab} \cong \tau_{bc} \cong \tau_{cd} > \sigma_{al} > \sigma_{ai}$ | Al indentation zone | Al base metal | - |

In case 1 of Table 3, the vibrational energy is transferred from the horn to the anvil. In this process, while the four base materials overlap, the two intermediate materials play the role of attenuation of energy transfer. Therefore, it was confirmed that the weldability of the overlapping materials was in the order of weld 1 > weld 2 > weld 3. In fact, as a result of the shear tensile test, as shown in Figure 8a, it was confirmed that the weld zone was fractured in the order of 3–2–1. Case 2 is the breaking sequence shown in Figure 8b, and case 3 is the breaking pattern of Figure 8c,d. In addition, in case 4, the strengths of the weld zones 1, 2, and 3 were higher than that of the aluminum base material, and it was fractured at the position of the aluminum base material.

### 3.1.3. Strength Curve

The type and sequence of fractures in the weld zone were mentioned above. This was confirmed by the shear tensile displacement–strength curves obtained from the actual shear tensile test. As was stated previously, when the ultrasonic multi-layer welding specimens are simultaneously tensioned, they have five types of displacement–strength curves according to the fracture type. Figure 10 shows the five typical types of displacement–strength curves obtained from this shear strength test. The E-curve in Figure 10 is the displacement–strength curve of the test specimen of the aluminum indentation fracture of weld zone 1 and the aluminum base material fracture between weld zones 2 and 3. In this case, the strength of the indentation fracture is insignificant compared to the strength of the aluminum base material, so it is impossible to identify the strength of the indentation through the curve. However, the C-curve represents specimen 7, exhibiting the interface fracture type of all weld zones. In this case, the strength values of each weld zone can be identified through the graph. The maximum weld zone tensile strength of C was very small compared to A. In case C, it is possible to identify the strength of each weld from the graph because the strength of each weld was similar.

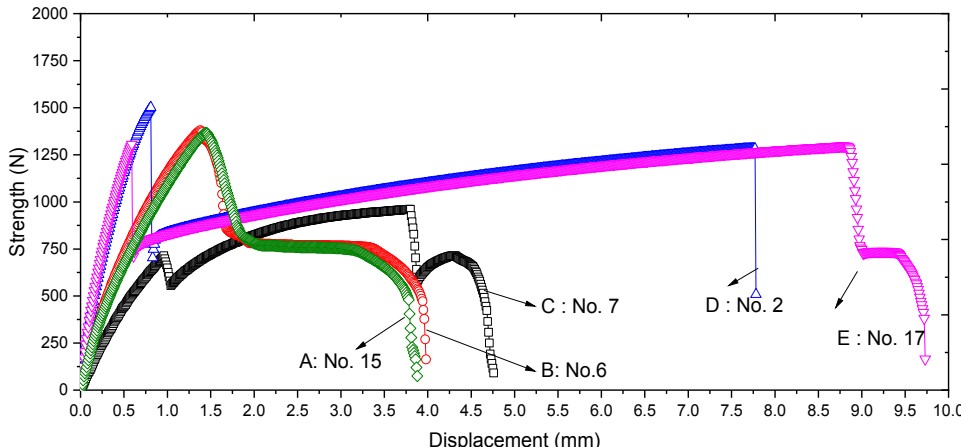

**Figure 10.** Five kinds of displacement–strength curves represented by multi-layer ultrasonic welding in a one-step shear strength test.

The individual displacement–strength curves of the welds are shown in Figure 11. As mentioned above, weld zone 1 shows three fracture types and weld zone 2 shows two fracture types. Weld zone 3 shows the interface fracture type, and Figure 11 shows the characteristics of the displacement–strength curve according to each fracture type.

The strength of each weld was compared by measuring the individual weld strength. Figure 12 compares the strength of each weld using the average of the strength values measured in the 25 welding conditions. The strength of the weld zone was the highest in weld zone 1. Weld zone 1 was where the horn came into direct contact with the aluminum surface, and received an indentation fracture, an aluminum base-material fracture, and an interface fracture. In weld zone 3, the strength was the lowest, at an average of 521.96 N. The average strength of weld zone 1 was 1,355 N, and the strength distribution was more than 1200 N in most parts of weld zone 1 because the dispersion width was very small. However, in weld zone 2, the average strength was 796.84 N and the distribution of the maximum and minimum value was very wide. In addition, the distribution range of strength was also very wide in weld zone 3. The influence of a change in welding conditions on strength was very small in weld zone 1, whereas weld zones 2 and 3 were very sensitive to the welding conditions. Although the melt area was not measured directly, the melt area could be predicted because the magnitude of the tensile load was relatively proportional to the size of melt area, as shown in Figure 12.

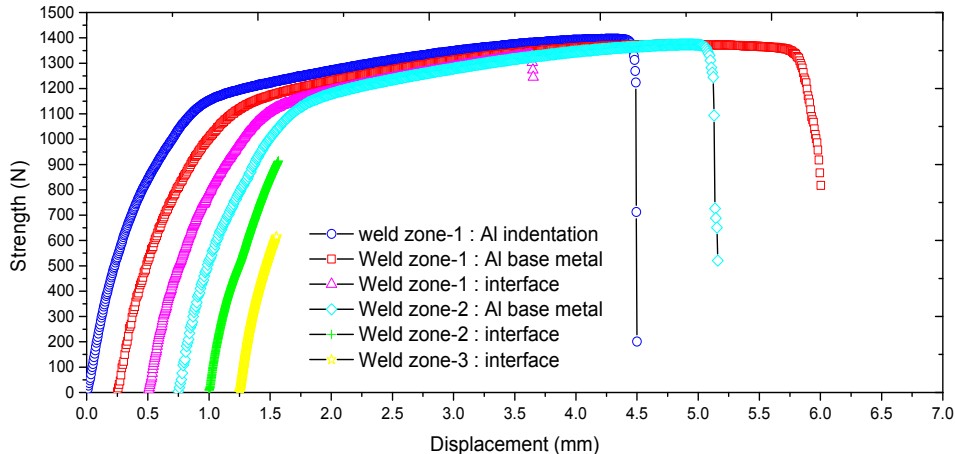

**Figure 11.** Six kinds of displacement–strength curves represented by multi-layer ultrasonic welding in a multi-step shear strength test.

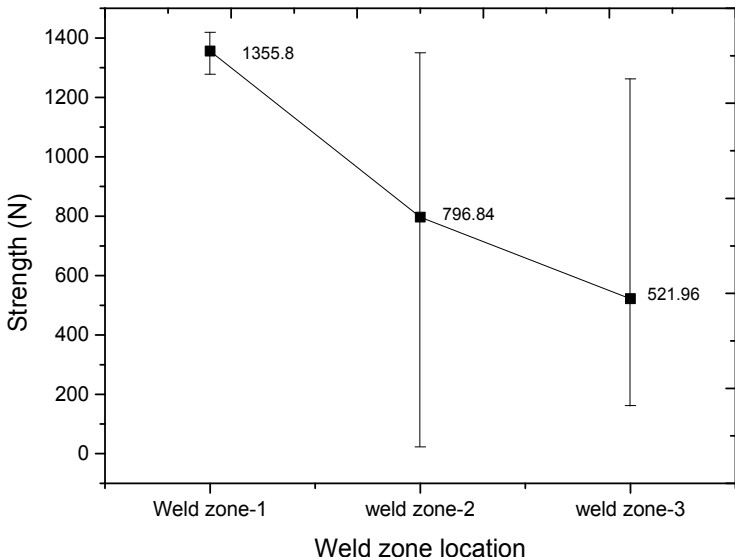

**Figure 12.** Average strength according to weld zone location.

Fractures can be broadly classified into three general types, according to their location: aluminum base material fracture, aluminum indentation fracture, and interface fracture. Strength according to each fracture type is compared in Figure 13. The aluminum base material fracture shows the strength of the aluminum material (1200 N or more), and the indentation fracture shows similar fracture strength to aluminum. In the case of an interface fracture, weld zone 1 shows a similar fracture strength to aluminum, but weld zones 2 and 3 show less than 1/2 the strength of weld zone 1. In terms of the frequency of the weld fracture, weld zone 1 showed an aluminum base material fracture type in 19 out of 25 experiments, and showed indentation and interfacial fracture types three times, respectively. Weld zone 2 showed an interface fracture type about 88% of the time. However, weld zone 3 exhibited interface fracture type in all of the 25 experiments. The interface fracture of weld zone 3 was found to be the lowest, at 521 N.

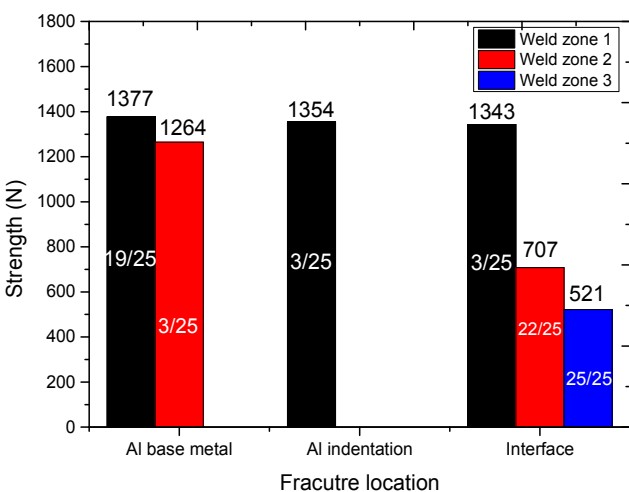

**Figure 13.** Average strength values according to fracture location.

Ultrasonic welding is performed by inserting material to be welded between an anvil and a horn and supplying pressure and vibrations. The vibrational energy is generated in the horn and transferred to the anvil, and the metal material inserted between the two devices acts as a damper, absorbing the vibration energy. The vibrational energy is mostly transferred closest to the horn in weld zone 1, and is changed into frictional energy. As the spring constant of the material acting as vibrational damping is proportionally to the square of the length, the farther away from the horn, the greater the amount of lost vibrational energy, causing the frictional energy to decrease [28,29]. Relatively, weld zone 3 has less frictional energy than weld zone 1, and an interface fracture occurs due to the cold-weld of the interface zone. In other words, for weld zone 3, away from the horn is not fully welded and an interface fracture occurs; the adjacent weld zone 1, in the horn is fully welded and a fracture occurs in the base metal. For this reason, most of the weld zone 3 has interface fractures.

### 3.2. The Optimization of Welding Conditions

Based on the S/N ratio analysis for each factor level designed in the experiment to determine the optimal welding conditions, the highest S/N ratio was measured at level 5 (80%) in factor A, level 4 (3.5 sec) in factor B (welding time), and level 0 (1.0 bar) in factor C (pressure load), so the optimal welding conditions were obtained at level $A_5B_4C_0$.

The S/N ratio calculated should be examined to ensure the reliability of the optimal welding conditions derived. As such, it is necessary to perform a variance analysis for the S/N ratio of the larger-the-better characteristics and to verify the significance of the welding variables that affect the strength of the weld. The variance analysis general linear model tool of the commonly used statistical program Minitab was used to perform the variance analysis of the S/N ratio of the larger-the-better characteristics, and the results are as shown in Table 3. The tensile strengths used for the S/N ratio calculations were taken from the simultaneous tensile strength results. Simultaneous tensile strength represents the strength of each weld joint, and the strengths of weld zones 1, 2, and 3 are combined to represent the maximum tensile strength.

The variance analysis found that the F-test is possible for all of the factors because the sum of the degrees of freedom of factors A, B, and C is 12, and the are a total of 24 degrees of freedom. Since there are no factors with an F-value less than 1, the significance level (*p*-value) of each factor was evaluated without a separate pooling process. Factors B and C were significant at the significance level of 5% (95% confidence interval) with *p*-values of 0.004 and 0.056, respectively, and A was not significant at the significance level of 5% with a *p*-value of 0.343, but the difference in the level is not negligible. Therefore, to obtain the optimal welding conditions, it is necessary to calculate the estimated value of the S/N ratio average for each factor level, and the highest S/N ratio average value according to each

factor level should be determined as the optimal welding condition. The average value of the S/N ratio for each factor level is calculated using the S/N ratio value of orthogonal arrays shown in Table 4, and Table 5 shows the S/N ratio average estimated value for each factor level.

**Table 4.** ANOVA (Analysis of Variance) for the signal to noise ratio (S/N) ratio of shearing strength.

| Factor | DF | SS | MS | F | P |
|---|---|---|---|---|---|
| A (Amplitude) | 4 | 123.54 | 30.89 | 1.25 | 0.343 |
| B (Welding time) | 4 | 663.53 | 165.88 | 6.7 | 0.004 |
| C (Pressure load) | 4 | 310.06 | 77.51 | 3.13 | 0.056 |
| Error | 12 | 296.94 | 24.74 | - | - |
| Total | 24 | 1394.07 | - | - | - |

**Table 5.** Welding parameters and conditions corresponding to the factors and levels.

| Factor | Level | Average S/N Ratio | Delta | Effect |
|---|---|---|---|---|
| A (Amplitude) | 0 | 49.99130 | 6.16 | 3 |
| | 1 | 55.82748 | | |
| | 2 | 53.86944 | | |
| | 3 | 53.05924 | | |
| | 4 | 56.15384 | | |
| B (Welding time) | 0 | 43.60410 | 13.55 | 1 |
| | 1 | 56.99988 | | |
| | 2 | 54.87088 | | |
| | 3 | 57.15572 | | |
| | 4 | 56.27072 | | |
| C (Pressure load) | 0 | 58.74044 | 9.33 | 2 |
| | 1 | 56.83480 | | |
| | 2 | 50.87994 | | |
| | 3 | 49.40824 | | |
| | 4 | 53.03788 | | |
| Total average | | 53.78026 | | |

Through calculating the S/N ratio average estimated value for each level, it was determined that the levels with the highest S/N ratio average estimated value were level 4 for factor A, level 3 for factor B, and level 0 for factor C. Table 5 summarizes the values corresponding to the welding experiment variables and welding conditions corresponding to each factor and level. Weld strength, the characteristic value of this study, is a larger-the-better characteristic, which is considered to be good if the S/N ratio is high, so the S/N ratio was calculated based on a combination of factors and levels corresponding to the optimal welding conditions using the Taguchi result prediction tool of the Minitab program.

The S/N ratio calculated by applying the optimal welding conditions resulted in a high value of 64.4985, shown in Table 6. Weld strength, the characteristic value of this study, is a larger-the-better characteristic, which is considered to be good if the S/N ratio is high; therefore, when compared to the S/N ratio value according to the experiment number in the L25(53) table of orthogonal arrays, the weld strength, as a larger-the-better characteristic, was determined to be the best, as the S/N ratio is highest under the optimized welding conditions.

**Table 6.** Optimal welding conditions and S/N ratio.

| Amplitude (%) | Welding Time (sec.) | Pressure Load (bar) | S/N Ratio |
|---|---|---|---|
| 80 | 3.5 | 1.0 | 64.4895 |

### 3.2.1. The Effects of Welding Parameters

Welding conditions, which are welding time, pressure load, and amplitude, effect weld strength, which is related to the fracture type and sequence. When the welding conditions are appropriate, the strength of weld zone is higher than that of base material and the fracture type is shown as Case 4 in Table 3. The main effect between the weld strength and the welding parameter was analyzed by the S/N ratio which represents the weld strength. The term delta in Table 5 is the slope of the S/N ratio according to the change of the relevant welding condition for each parameter. The maximum delta value of welding time is 13.55, and the influence on the weld strength is larger than other parameters. It is effective to adjust welding time, pressure load, and amplitude in that order, for securing the desired strength through the adjustment of welding conditions.

### 3.2.2. Comparison of the Welding Quality

Additional experiments were carried out to evaluate the performance of weld zone under optimum conditions. The non-optimizing welding conditions for the comparison with the optimized welding conditions were selected by the highest welding strength among the 25 tests performed previously. Non optimizing welding conditions, such as 50% of amplitude, 3.5sec of welding time, and 1.0 of pressure load were used. The experimental method was divided into two parts; first, the maximum strength was compared when both the aluminum plate and the copper plate were simultaneously tensioned. In addition, three shear strength measurement tests were performed for each specimen to determine the strength of the welds individually. Figure 14 shows the displacement–strength curve when all of the welding specimens were tensioned using optimized welding conditions and non-optimized experiment conditions. Both welding conditions exhibited similar displacement–strength curves. The maximum tensile strength was confirmed to be 45% higher under optimized welding conditions compared to other cases. The maximum tensile strength refers to the load which the two sheets of aluminum can withstand. This is represented by the load applied to the entire material, when the four plates are overlapping. This refers to the force before the fracture of the first aluminum indentation. Afterwards, the second aluminum plate withstands all tensile loads, causing the aluminum base material to fracture. The fracture occurs due to the plastic deformation of the second aluminum, and the section where the load is maintained consistently during the constant displacement is the section where the second aluminum is deformed.

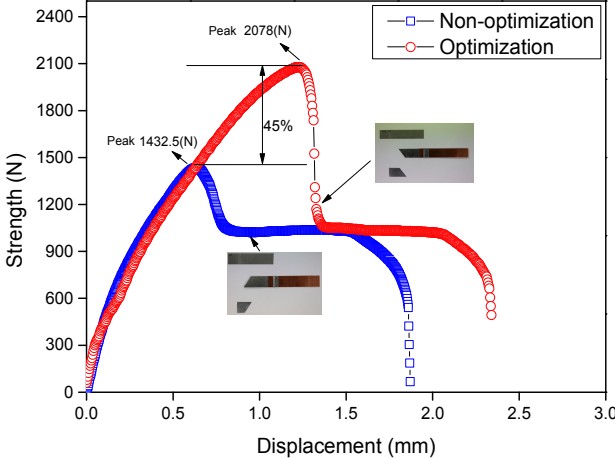

**Figure 14.** Comparison strength between non-optimized and optimized conditions.

Figure 15 is a graph measuring the shear load of each weld zone. Through identifying the strength of each weld, it was found that when the optimal conditions are not used, the weldability decreased as the distance of the weld from the horn increased, and the distribution of the strength levels becomes very large. However, when the optimal conditions were used, the strength was reduced in the

aluminum base material directly in contact with the mixed material, but the strength was improved for weld zones 2 and 3. Weld zone 1 showed a reduction in strength of about 36%, but weld zone 2 showed a strength improvement of 32%, while weld zone 3 showed a strength improvement of 110%. As an amplitude rate of 80% was used, that was considered to be a result of the decrease in the thickness of the indentation as the depth of the indentation on the surface becomes deeper. Figure 16 shows the thickness of the aluminum weld according to the depth of the indentation. The depth of the indentation was 270 μm when the optimal welding conditions were used, and was about 80 μm thinner under normal welding conditions. As the amplitude increased, the size of the indentation of aluminum surface increased and the bonding strength of the weldment increased, but that had the effect of reducing the thickness of occured the indentation. In addition, as shown in Figure 16, since the fracture position in the weld 1 at the aluminum indentation, the reduction of the indentation thickness was directly related to the strength reduction. Nevertheless, the strength of welds 2 and 3 was improved. That has the effect of improving the overall strength of the welded part.

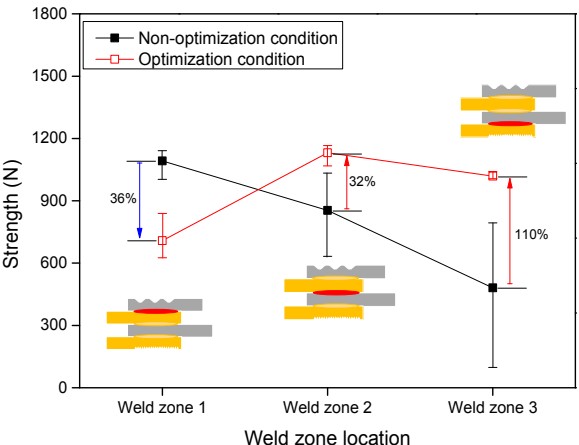

**Figure 15.** Comparison of strength between non-optimized conditions and optimized conditions at each weld zone.

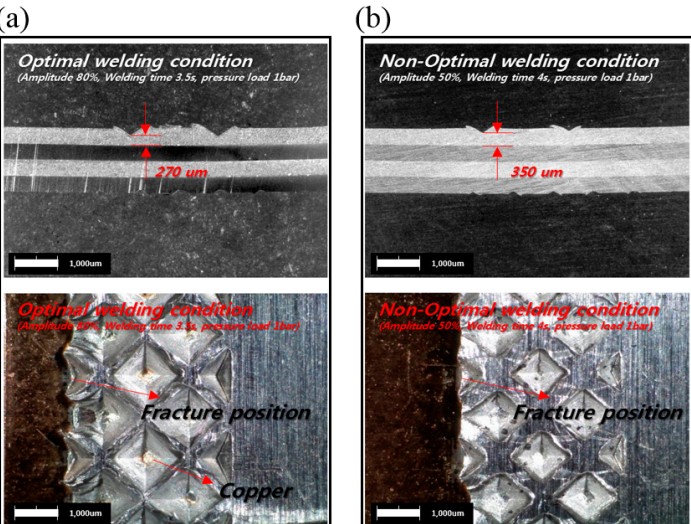

**Figure 16.** Comparison of cross section for aluminum indentation thickness between optimal welding conditions and non-optimal welding conditions. (**a**) Optimal welding conditions; (**b**) non-optimal welding conditions.

### 3.2.3. Analysis of the Interfacial Shape

In the ultrasonic welding of aluminum and copper sheets, tearing occurs in aluminum because copper is harder. Looking at the two interfaces, there was an indentation pattern of a circle on the opposite side of the aluminum. Both interfaces showed vibration marks in the transverse direction, and the aluminum tissue was observed on the copper interface, as shown in Figure 17b. The distribution of aluminum on the copper interface was observed to determine the cause of the difference in weld strength between the case with and without the optimized conditions.

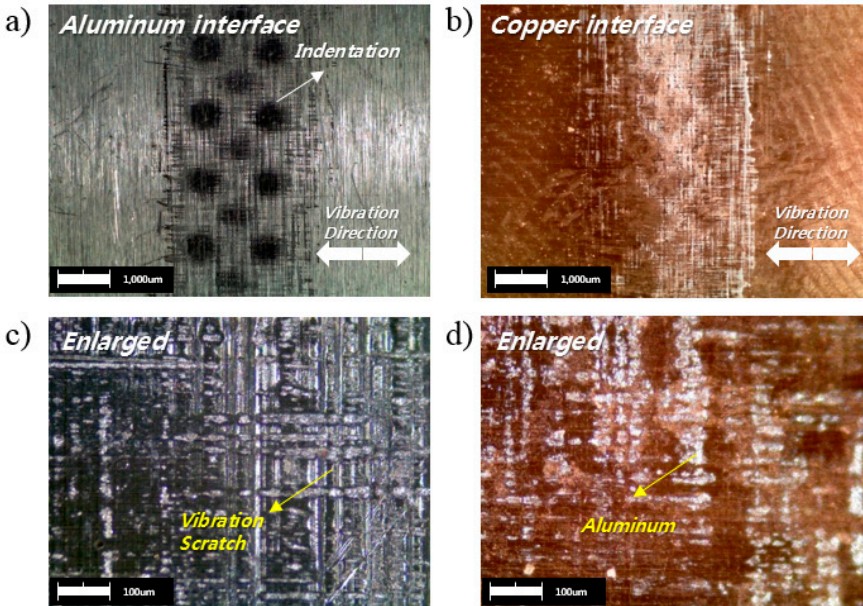

**Figure 17.** Comparison of the interface shape of Al and Cu: (**a**) aluminum interface, (**b**) copper interface, (**c**) enlarged aluminum interface, and (**d**) enlarged copper interface.

Figure 18 presents an observation of the degree to which aluminum settled on the copper plate using the contrast ratio difference depending on the mass of the tissue component. The settlement amount of aluminum cannot be identified quantitatively, but the approximate amount of settlement can be determined by observing the distribution of aluminum in the same area. After the shear tensile test, the surface was observed by SEM and the distribution of the compressed aluminum on the surface of the copper sheet was confirmed. It was observed in the order of x100, x200, and x500 times for surface observation. The settlement amount is related to the welding performance of the two metals. The greater the settlement amount, the better the bonding performance, which leads to a strength improvement. By comparing the settlement amount according to the two welding conditions, it was found that the distribution size of aluminum was larger when the optimized welding conditions were used compared to when they were not used. In addition, when compared according to the location of the weldment, weld zone 2, which was close to the indentation, was confirmed to have a larger aluminum distribution compared to weld zone 3.

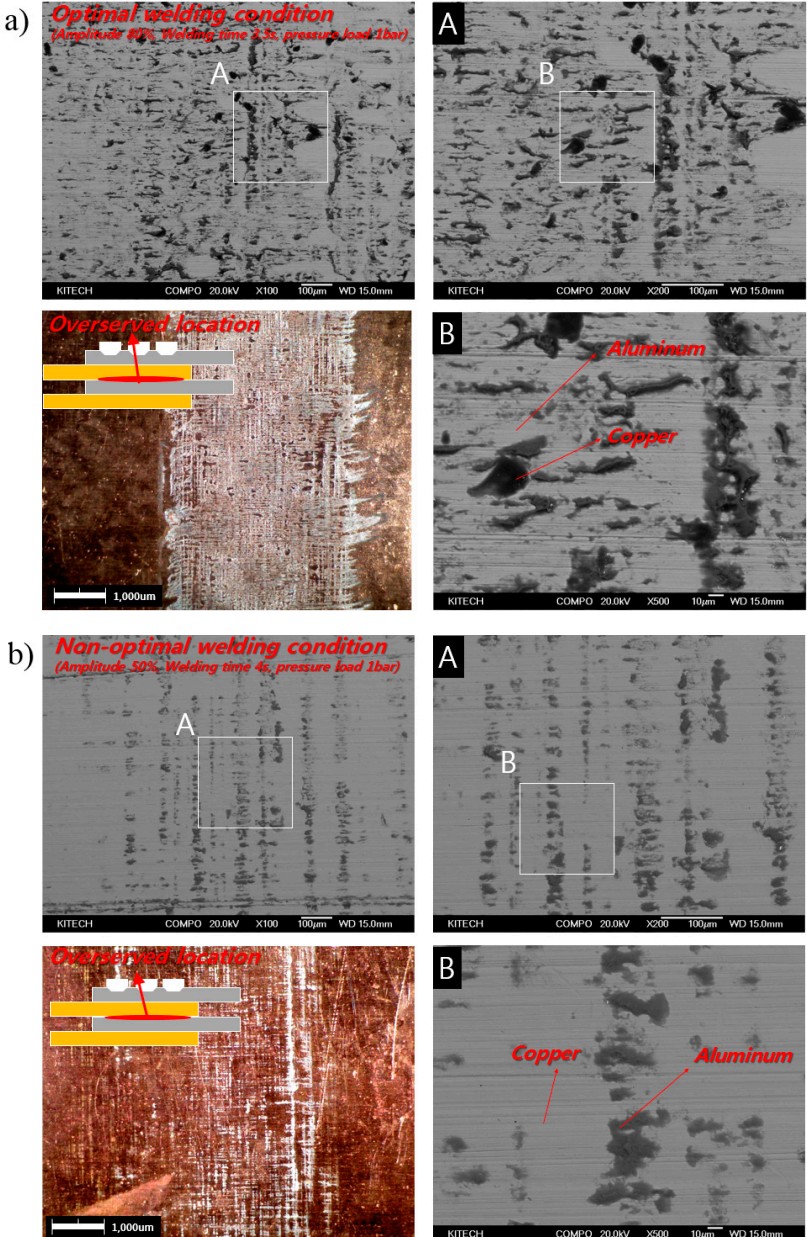

**Figure 18.** SEM images of fractured weld zone on the Cu surface with optimal welding conditions and non-optimal welding conditions: (**a**) optimal welding conditions, and (**b**) non-optimal welding conditions.

## 4. Conclusions

In this study, the multi-layer ultrasonic welding characteristics of aluminum and copper sheets were analyzed, and the strength, fracture type, and interface were observed according to the weld location. Multi-layer ultrasonic welding conditions were optimized using the Taguchi method, and the detailed results are as follows.

(1) To optimize the quadruple lap ultrasonic welding process conditions of 0.4 t aluminum and copper sheets, an experimental plan was established using the Taguchi method, and the experiment was performed with 25 conditions. For strength evaluation, the ultrasonic welding performance was evaluated by measuring the tensile strength as a composite material and the shear force at the weld interface through two types of tensile tests: simultaneous tensile and individual tensile.

(2) Using the tensile test of the weld as a composite material through simultaneous tensile tests, it was verified that there were five types of fracture, which were caused by the difference of stress in each weld. In the simultaneous tensile test, the displacement–strength curve showed the yield point at two positions, which was caused by a series of fractures of the aluminum base material.

(3) To identify the individual shear strengths of the multi-layer, dissimilar ultrasonic welds, three tensile tests were performed for each specimen depending on the location of the weld, and it was found that as the distance from the horn increased, each of the shear strengths decreased, while the difference in strength values increased.

(4) The Taguchi S/N ratio was used to optimize the welding conditions, and strength was selected as the characteristic value to optimize based on its 'larger-the-better' characteristics. For quadruple lap welding of pure aluminum and copper sheets (0.4t), the S/N ratio was the highest at 64.48 when the pattern was—Pitch: 1.5 mm, Height: 0.75 mm, and Stub tooth: 0.70mm; and welding conditions were—amplitude: 80%, welding time: 3.5s, pressure load: 1.0 bar. Those were selected as the optimal conditions.

(5) To evaluate the optimized welding conditions, additional tests were conducted using the welding conditions showing the maximum strength values and the optimized welding conditions using the Taguchi method through simple tests. A strength evaluation of the optimized weld was performed, and for the simultaneous tensile test, it was found that the strength of the optimized weld was improved by 45% compared to other cases.

**Author Contributions:** Conceptualization, I.-j.K.; methodology, S.K. and K.C.; formal analysis, J.K. (Jisun Kim), J.K. (Jaewoong Kim).

**Funding:** This study has been conducted with the support of the Korea Institute of Industrial Technology as "Study on the Optimal Welding Condition of Titanium Materials for Manufacturing Ship Scrubber (PJA19300)".

**Conflicts of Interest:** The authors declare no conflict of interest.

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
