# Peer review of "An Analysis of Mechanical Properties for Ultrasonically Welded Multiple C1220-Al1050 Layers"

_applsci, doi:10.3390/app9194188_

Round 1

Reviewer 1 Report

I highly suggest for you to strengthen the literature review.

For example:

line 56 : Ultrasonic welding has  recently been applied to bonding nickel and aluminum in automobile battery cells or micro-wire  bonding of electronic products.

No reference was given.

Only three references give the idea that this type of welding is really not an important issue these days and its use despicable.

It is recommended for you to show a cross-section of the interface before applying the loads. No figure was given regarding samples in the as-built condition.

Author Response

## Reviewer 1

Reviewer comments (1) : I highly suggest for you to strengthen the literature review

Answers and corrections: The literature was added to increase the credibility of the paper. Corrected content has been inserted on paper. (Line 35 ~ 62), (line 492~549, Reference)

Reviewer comments (2) :  It is recommended for you to show a cross-section of the interface before applying the loads. No figure was given regarding samples in the as-built condition

Answers and corrections: The relevant content has been inserted in the paper. (Line 186 ~ 196)

Figure 7. Cross-section ultrasonic weld zone with amplitude 70%, welding time 2.5sec, pressure load 3 bar; (a) Cross section cutting position, (b) Joint cross section (x40), (c) Joint cross section (x3000)

The shape of the welded specimen is shown Figure 7(a). At the cross section in Figure 7(b), bond line was not observed due to the characteristics of solid phase bonding. When resistance welding, the bond line is observed but both heat affected zone and bonding line are not observed since the ultrasonic metal welding is one of the plastic flow welding method [26.27]. However, it is confirmed that two metals were closely bonded to each other after observing it by 3000 times magnification as shown Figure 7(c).

The relevant content has been inserted in the paper. (Line 285 ~ 287)

Although the melt area was not measured directly, the melt area can be predicted because the magnitude of the tensile load is relatively proportional to the size of melt area as shown in Figure. 12.

Reviewer 2 Report

This is an interesting experimental study, well documented in this paper. The work has publication merit. Some notes:

The abstract needs to be shortened, removing dimensional references. Fig. 7 should be more clear  (larger font size) - the same applies for Fig. 8 Fig. 12: correct 'fracutre' in the caption Table 4 should not span across two pages Scale indicators (markers) are missing from the images shown in Fig. 16, 17 and 18

Author Response

## Reviewer 2

Reviewer comments (1) : The abstract needs to be shortened, removing dimensional references

Answers and corrections: Comments are reflected in Abstract as shown below.

This study analyzed the characteristics of aluminum and copper sheets under multi-layer ultrasonic welding, and observed the strength, fracture type, and interface of the weld zone according to location. In addition, an experiment plan was developed using the Taguchi method to optimize the quadruple lap ultrasonic welding process conditions of 0.4t aluminum and copper sheets, and the experiment was performed twice with 25 times each. For strength evaluation, the ultrasonic welding performance was evaluated by measuring the tensile strength as a composite material and the shear force at the weld interface through two types of tensile tests: simultaneous tensile and individual tensile. To identify the individual shear strengths of the multi-layer dissimilar ultrasonic welds, three types of tensile tests were performed for each specimen depending on the location of the welded, and as the distance from the horn increased, each of shear strength decreased while the difference in strength value increased. For quadruple lap welding of pure aluminum and copper sheets, the S/N ratio was the highest at 64.48 with Coarse-grain pattern and optimal welding conditions, and this was selected as the optimal condition. To evaluate the optimized welding condition, additional tests were conducted using the welding conditions that showed the maximum strength values and the welding conditions optimized using the Taguchi method through simple tests. A strength evaluation of the optimized weldment was performed, and for simultaneous tensile, it was found that the strength of the optimized weldment was improved by 45 % compared to other cases.

Reviewer 3 Report

This paper implements some interesting tests to characterize the mechanical property of ultrasonic welded multiple Cu/Al layers. Some revisions are required before the acceptance of this paper.

When the multi-layer ultrasonic welds are simultaneously tensioned, are the three weld zones fractured at the same time or in sequence? Fig.7 shows the fracture types at one step shear strength test. How the welding parameters influence the fracture type? From line 287 to line 308, there are some discussions on FSW and inherent strain method. It seems that these contents have no relationship with the topic of this paper. There is no Table 5 in the PDF file. Which weld strength, the simultaneously tensioned one or the individually tensioned one, was used to calculate the S/N ratio? Cross-section of the welded joint should be provided to show the bonding line at each weld zone.
